# Impact of Advanced Age on the Incidence of Major Adverse Cardiovascular Events in Patients with Type 2 Diabetes Mellitus and Stable Coronary Artery Disease in a Real-World Setting in Spain

**DOI:** 10.3390/jcm12165218

**Published:** 2023-08-10

**Authors:** Carlos González-Juanatey, Manuel Anguita-Sánchez, Vivencio Barrios, Iván Núñez-Gil, Juan José Gómez-Doblas, Xavier García-Moll, Carlos Lafuente-Gormaz, María Jesús Rollán-Gómez, Vicente Peral-Disdier, Luis Martínez-Dolz, Miguel Rodríguez-Santamarta, Xavier Viñolas-Prat, Toni Soriano-Colomé, Roberto Muñoz-Aguilera, Ignacio Plaza, Alejandro Curcio-Ruigómez, Ernesto Orts-Soler, Javier Segovia-Cubero, Víctor Fanjul, Judith Marín-Corral, Ángel Cequier

**Affiliations:** 1Hospital Universitario Lucus Augusti, 27003 Lugo, Spain; 2Instituto Maimonides de Investigación Biomédica de Córdoba (IMIBIC), Hospital Universitario Reina Sofía, Universidad de Córdoba, 14014 Cordoba, Spain; manuelanguita@secardiologia.es; 3Hospital Universitario Ramón y Cajal, 28034 Madrid, Spain; vivenciobarrios@gmail.com; 4Cardiology Department, Hospital Clínico Universitario San Carlos, 28040 Madrid, Spain; ibnsky@yahoo.es; 5Faculty of Biomedical and Health Sciences, Universidad Europea de Madrid, Villaviciosa de Odón, 28670 Madrid, Spain; 6IBIMA (Instituto de Investigación Biomédica de Málaga), Hospital Universitario Virgen de la Victoria, CIBERCV (Centro de Investigación Biomédica en Red Enfermedades Cardiovasculares), 29010 Malaga, Spain; jjgomezdoblas@gmail.com; 7Hospital Universitario Santa Creu i Sant Pau, 08041 Barcelona, Spain; xgarcia-moll@santpau.cat (X.G.-M.); xvinolas@santpau.cat (X.V.-P.); 8Hospital Universitario de Albacete, 02006 Albacete, Spain; carloslafuentegormaz@gmail.com; 9Hospital Universitario Río Hortega, 47012 Valladolid, Spain; mariajrollan@yahoo.es; 10Hospital Universitario Son Espases, 07120 Palma de Mallorca, Spain; vicente.peral@ssib.es; 11Hospital Universitario y Politécnico La Fe, CIBERCV (Centro de Investigación Biomédica en Red Enfermedades Cardiovasculares), IIS La Fe, 46026 Valencia, Spain; martinez_luidol@gva.es; 12Hospital Universitario de León, 24071 Leon, Spain; mrsantamart@gmail.com; 13Hospital Vall d’Hebron, CIBERCV (Centro de Investigación Biomédica en Red Enfermedades Cardiovasculares), 08035 Barcelona, Spain; tonisorianocolome@gmail.com; 14Hospital Infanta Leonor, 28031 Madrid, Spain; rmunoza@salud.madrid.org; 15Hospital Infanta Sofía, 28703 Madrid, Spain; ignacioplazap@gmail.com; 16Hospital Universitario de Fuenlabrada, 28942 Madrid, Spain; alejandro.curcio@salud.madrid.org; 17Hospital General Universitario de Castellón, 12004 Castellon de la Plana, Spain; eorts@comcas.es; 18Hospital Universitario Puerta de Hierro, 28222 Madrid, Spain; jsecu@telefonica.net; 19Savana Research SL, 28013 Madrid, Spain; vfanjul@savanamed.com (V.F.); jmarin@savanamed.com (J.M.-C.);; 20Hospital Universitario de Bellvitge, IDIBELL (Instituto de Investigación Biomédica de Bellvitge), Universidad de Barcelona, 08007 Barcelona, Spain; acequier@bellvitgehospital.cat

**Keywords:** type 2 diabetes mellitus, coronary artery disease, MACE, real-world data, electronic health records, natural language processing, aging

## Abstract

Patients with type 2 diabetes mellitus (T2DM) and coronary artery disease (CAD) without myocardial infarction (MI) or stroke are at high risk for major cardiovascular events (MACEs). We aimed to provide real-world data on age-related clinical characteristics, treatment management, and incidence of major cardiovascular outcomes in T2DM-CAD patients in Spain from 2014 to 2018. We used EHRead^®^ technology, which is based on natural language processing and machine learning, to extract unstructured clinical information from electronic health records (EHRs) from 12 hospitals. Of the 4072 included patients, 30.9% were younger than 65 years (66.3% male), 34.2% were aged 65–75 years (66.4% male), and 34.8% were older than 75 years (54.3% male). These older patients were more likely to have hypertension (OR 2.85), angina (OR 1.64), heart valve disease (OR 2.13), or peripheral vascular disease (OR 2.38) than those aged <65 years (*p* < 0.001 for all comparisons). In general, they were also more likely to receive pharmacological and interventional treatments. Moreover, these patients had a significantly higher risk of MACEs (HR 1.29; *p* = 0.003) and ischemic stroke (HR 2.39; *p* < 0.001). In summary, patients with T2DM-CAD in routine clinical practice tend to be older, have more comorbidities, are more heavily treated, and have a higher risk of developing MACE than is commonly assumed from clinical trial data.

## 1. Introduction

Type 2 diabetes mellitus (T2DM), which accounts for more than 90% of all cases of diabetes, is a major global health concern that places a heavy burden on the public health system and the socioeconomic development of all nations [1,2]. Due to the combined effects of aging, excess body weight, sedentary lifestyles, and unhealthy eating habits, among other factors,T2DM has reached epidemic proportions [3]. According to the International Diabetes Federation (IDF), the global prevalence of diabetes in 2021 was estimated at 9.8% (537 million cases), with a total diabetes-related healthcare expenditure exceeding USD 960 billion. These figures are expected to rise to 11.2% (784 million cases) and USD 1053 billion by 2045 [4].

T2DM is associated with the development of clinical complications that impair the functional capacity and health-related quality of life of patients, leading to significant morbidity and a twofold increase in mortality when compared with the general population [5,6,7]. People with T2DM are at a significantly higher risk of cardiovascular comorbidities than nondiabetic subjects, including myocardial infarction (MI), stroke, peripheral artery disease, and coronary artery disease (CAD) [8,9,10,11]. In this regard, T2DM elicits cell signaling, epigenetic, and posttranslational changes that directly or indirectly affect the biology of the vasculature and other metabolic systems, especially in the endothelium, liver, skeletal muscle, and β cells. This results in the development of cardiovascular disease (CVD) [12].

In a recent systematic review, the global prevalence of CVD among individuals with T2DM was estimated at 32.2%. Moreover, CVD was identified as the leading cause of mortality among T2DM patients, accounting for approximately half of all deaths during the study period. CAD was the major contributor, accounting for 29.7% of all CVD-related deaths [13]. The relative risk of morbidity and mortality from CAD among people with diabetes has been reported to be higher in women than in men [14,15,16]. Globally, the total number of deaths attributable to diabetes among the population aged 20–79 years was estimated at 6.7 million in 2021. Of these, 67.4% occurred in individuals aged ≥60 years. In Spain, 81,717 deaths due to diabetes were recorded in the same year, of which 98.3% occurred in patients over 60 years of age. Similar percentages were also recorded in other neighboring countries, such as France (98.5%), Germany (98.3%), Italy (98.2%), and Portugal (97.7%) [4]. In this sense, age is one of the main risk factors for CVD and is associated with the development of other CVD risk factors, including obesity and diabetes [7,17]. The most prevalent types of CVD in the elderly are CAD, heart failure with preserved ejection fraction, myocardial infarction, and arrhythmias such as atrial fibrillation [17].

Lifestyle interventions are a key first step in the management of patients with T2DM. However, most patients eventually need medication [18]. Until recently, the standard treatment for T2DM was mainly based on the use of glucose-lowering drugs [19]. However, owing to the uncertainty regarding the cardiovascular safety of these agents, the US Food and Drug Administration (FDA) and the European Medicines Agency (EMA) updated their guidelines, requiring the evaluation of all new glucose-lowering drugs in long-term cardiovascular outcome trials (CVOTs) [20,21]. A number of CVOTs evaluating the cardiovascular safety of new hypoglycemic agents have revealed clinical findings that are far greater than originally expected ([22] and references therein). These new data are leading to a paradigm shift in the management of T2DM, prioritizing sodium–glucose cotransporter 2 inhibitor (iSGLT2) treatment in patients with CAD because of its their cardioprotective role [23,24]. In this regard, geriatric patients with a higher risk of CVD may respond differently to drug therapy than younger patients [13,25]. However, elderly patients have often been excluded from cardiovascular trials, including those influencing current treatment guidelines, in many cases because arbitrary upper age limits are not medically justified, thereby compromising the external validity of trial findings [26,27,28,29,30]. In addition, few studies have quantified the effect of age on CVD prevalence among individuals with T2DM [13]. In this context, real-world data are becoming increasingly important to accelerate improvements in patient care.

Here, we present the results of a multicenter, retrospective, and observational study in which unstructured information from electronic health records (EHRs) was extracted and analyzed using natural language processing (NLP) and machine learning (ML). Our aim was to provide real-world data on age-related clinical characteristics, treatment management, and major adverse cardiovascular events (MACEs) in patients with T2DM and stable CAD in Spain.

## 2. Materials and Methods

### 2.1. Study Design

This study was a subanalysis of the real-world, multicenter, retrospective, and observational ACORDE study based on the secondary use of unstructured data captured in the EHRs [31]. We evaluated the age-related clinical characteristics, treatment management, and incidence of MACE in T2DM-CAD patients between 1 January 2014, and 31 December 2018. A cross-sectional analysis of all patients stratified according to age ranges previously described (<65 years, 65–75 years, and >75 years) was performed at index date [32]. Index date was defined as the first time in the study period that the patient fulfilled all the inclusion criteria and no exclusion criteria. At this point, demographics, comorbidities, vital signs, general characteristics of T2DM, CAD, and treatments were evaluated according to age group. The cumulative incidence of MACEs was analyzed during the follow-up by age group and defined by the presence of MI, stroke, hospitalization for unstable angina, or urgent coronary revascularization [33]. Although included in this definition, all-cause or cardiovascular death was not analyzed because the nature of the data source did not allow for their occurrence to be accurately identified. The follow-up period was defined as the period from the index date to the last EHR available for each patient during the study period.

### 2.2. Data Source

The data source was the free-text information within the EHRs of 12 representative hospitals from six major regions of Spain, including Madrid (Hospital Universitario de Fuenlabrada, Hospital Universitario Infanta Sofía, Hospital Universitario Infanta Leonor and Hospital Universitario Puerta de Hierro), Catalonia (Hospital Universitari Vall d’Hebron and Hospital de la Santa Creu i Sant Pau), Valencia (Hospital Universitari i Politècnic La Fe and Hospital General Universitario de Castellón), Balearic Islands (Hospital Universitari Son Espases), Castilla-La Mancha (Complejo Hospitalario Universitario de Albacete), and Castilla y León (Hospital Universitario Río Hortega and Hospital Universitario de León). Data on outpatient clinical reports, discharge reports, emergency reports, prescriptions, and other medical reports were collected from all available departments at each participating site, including the inpatient, outpatient, and emergency departments.

### 2.3. Study Population

The source population of the study comprised all adult patients with available EHRs in the participating hospitals during the study period. Patients with T2DM and a diagnosis of CAD were included. These two entities were included if they were present in unstructured free-text information in the EHRs, based on clinical diagnosis. T2DM was also considered if there was documented ongoing use of glucose-lowering medication (oral hypoglycemic agents) for at least 6 months. CAD was considered if there was evidence of stenosis ≥50% of at least one coronary artery, but without a history of previous MI or stroke, and without planned coronary, cerebrovascular, or peripheral arterial revascularization. Patients with prior MI or stroke, history of liver cirrhosis or liver cancer, intracranial bleeding, renal failure requiring dialysis, ongoing treatment with anticoagulant medication at index date, or unavailable follow-up information spanning at least six months were excluded from the study. Some of the exclusion criteria were aimed at better selection of the incident CAD population during the study period.

### 2.4. Extraction of Clinical Data from EHRs

Anonymized clinical information was extracted from the EHRs of participating hospitals. Data on date of birth and sex were extracted from structured data and age was computed at index. All other variables in the study, including clinical characteristics, analytical parameters, comorbidities, pharmacological treatments, interventional procedures, and specific MACE, were extracted from unstructured clinical data using the EHRead^®^ (MedSavana, Madrid, Spain). This technology uses NLP and ML techniques to extract free text from deidentified and processed EHRs and translate it into a study database [34,35,36,37,38,39,40,41,42]. The terminology considered by EHRead^®^ included codes, concepts, synonyms, and definitions used in clinical documentation and was based on SNOMED CT [43,44]. The performance of EHRead technology was evaluated and detailed in earlier articles of this study [34,35].

### 2.5. Statistical Data Analyses

Descriptive tables were generated to show the distribution of demographic, clinical, and treatment characteristics, as well as the development of MACEs by age group (i.e., <65, 65–75, and >75 years). Categorical and binary variables were presented as frequencies, whereas numerical variables were summarized as mean and standard deviation (SD) or median and interquartile range (IQR), as specified. Frequencies of available or missing values were also reported. Missing information for binary variables was interpreted as no occurrence or absence of the characteristic (i.e., true zero values). No further imputation of the missing data was applied. To statistically compare age groups in terms of categorical variables, we tested the null hypothesis (equal proportions) using logistic regression models. To compare the age groups with respect to numerical variables, we tested the null hypothesis (equal means) using linear regression models. Kaplan–Meier (KM) analysis and Cox proportional hazards (PH) regression models were used for time-to-event analysis (MACE development). MACEs during the follow-up period were considered an event, and patients who had no MACE during the follow-up period were censored. Differences were considered statistically significant if *p* < 0.05 in two-tailed tests. *p*-values were adjusted using the Benjamini–Hochberg method when accounting for multiple hypothesis testing. The KM curves were displayed as plots. Data were analyzed and represented using R software v4.0.2 (The R Foundation for Statistical Computing, Vienna, Austria).

### 2.6. Ethical Considerations and Study Approval

This study was classified as a non-post-authorization study by the Spanish Agency of Medicines and Health Products (AEMPS) and was approved by the Institutional Review Board (IRB) of each participating site. All methods and analyses were performed in compliance with local legal and regulatory requirements as well as the generally accepted research practices described in the latest version of the Declaration of Helsinki and Good Pharmacoepidemiology Practices. Data were analyzed from deidentified EHRs, which were aggregated in an irreversibly dissociated manner. Therefore, individual patient consent was not required in the study.

## 3. Results

### 3.1. Study Population

A total of 2,184,662 EHRs, containing clinical data from 217,632 patients, were processed in 12 participating hospitals during the study period. Of all the patients, 4072 were adults, had a diagnosis of both T2DM and stable CAD, with no previous history of MI or stroke, and with a minimum follow-up information of 6 months. These constituted the study population [31]. By age group, 1260 (30.9%) patients were younger than 65 years, 1393 (34.2%) were aged 65–75 years, and 1419 (34.8%) were older than 75 years (Figure 1). Patients were followed up for a median of 33.6 months (IQR, 19.5 to 47.1). T2DM and CAD were first reported in the EHRs of included patients at a median age of 67 (IQR, 58 to 75) and 69 (IQR, 60 to 76) years, respectively.

### 3.2. Patient Demographic and Clinical Characteristics

The baseline demographic and clinical characteristics of the patients in both the study population and by age group are shown in Table 1. The male-to-female ratio was 1.6 (62.2% vs. 37.8%). However, according to age group, the relative proportion of men was lower in patients older than 75 years (54.3%) than in those aged 65–75 (66.4%; odds ratio (OR) 0.60; *p* < 0.001) and in those younger than 65 years (66.3%; OR 0.61; *p* < 0.001). Overall, 54.2% of the patients were current or former smokers, with the proportion of active smokers decreasing significantly with increasing age (*p* < 0.001 for all comparisons). The specific type of CAD was detected in 60% of the patients, of whom 20.0% had single coronary vessel disease, 39.3% had multivessel coronary disease, and 0.7% had left main disease. The percentage of patients with a known type of CAD increased with age: 53% of patients were younger than 65 years (17.1% single-vessel and 35.2% multivessel CAD), 60.6% of patients were aged 65–75 years (19.8% and 40.3%), and 65.5% of patients were older than 75 years (22.6% and 42.1%, respectively). Some clinical parameters were consistent with the presence of metabolic abnormalities, including a BMI within the obesity range (median 30.5 kg/m^2^; IQR, 26.9 to 35.4) and elevated glucose levels (median 135 mg/dL; IQR, 113.0 to 168.0). However, BMI was significantly higher in patients aged <65 years than in those aged >75 years (median 31.6; IQR, 27.4 to 37.0 vs. 29.3; IQR, 26.3 to 32.0; β = –2.17; *p* = 0.006). Among the most common baseline comorbid conditions, the frequency of CVD showed a significant increasing trend with advancing age (Table 2). Patients aged >75 years were more likely to have arterial hypertension (OR 2.85), angina (OR 1.64), heart valve disease (OR 2.13), and peripheral vascular disease (OR 2.38) than those aged < 65 years. Conversely, they were less likely to have hyperlipidemia (OR 0.73) and less likely to be obese (OR 0.38) (*p* < 0.001 for all comparisons, Table 2).

### 3.3. Pharmacological and Interventional Disease Management

In general, older patients were more likely to receive pharmacological and interventional treatments than younger patients were (Table 3 and Table 4). There were two exceptions: patients older than 75 years were 1.37 times less likely to receive metformin (OR 0.73; *p* < 0.001), the most commonly prescribed oral hypoglycemic agent, and 1.33 less likely to receive insulin (OR 0.75; *p* = 0.002) than those younger than 65 years. In contrast, older patients were significantly more likely to receive sulfonylureas (OR 1.41, *p* < 0.001) and DPP4 inhibitors (OR 1.32, *p* = 0.004). Likewise, patients older than 75 years were significantly more likely to receive aspirin alone (OR 1.42) and clopidogrel (OR 1.40) than patients under 65 years of age (*p* < 0.001 for both comparisons). Most patients received statins to control hyperlipidemia, and their use increased with advancing age. Compared to those aged <65 years, patients aged >75 years were significantly more likely to receive statins (OR 1.33, *p* = 0.002). Pharmacological treatments for the management of arterial hypertension were also prescribed more commonly in older patients, including angiotensin-converting enzyme (ACE) inhibitors (OR 1.30; *p* = 0.001), angiotensin receptor blockers (ARBs) (OR 1.83; *p* < 0.001), and beta blockers (OR 1.28; *p* = 0.002). Regarding interventional procedures, 38.8% of the patients in the study population underwent percutaneous coronary intervention (PCI) and 14.4% underwent coronary artery bypass grafting (CABG). The use of CABG was significantly more common in the older age groups. Patients aged 65–75 years and those over 75 years were significantly more likely to undergo CABG than those aged <65 years (OR 1.40, *p* = 0.003 and OR 1.39, *p* = 0.004, respectively).

### 3.4. Cumulative Incidence of MACE

The likelihood of developing MACEs in patients with T2DM and stable CAD increased with advancing age. Older patients were significantly more likely to have at least one MACE, particularly an ischemic stroke. Compared with patients younger than 65 years of age, the estimated risk of MACEs after 48 months of follow-up increased by 18% in patients aged 65–75 years (*p* = 0.051) and by 29% in patients older than 75 years (*p* = 0.003). The corresponding risks for ischemic stroke in the same age group increased by 74% (*p* = 0.007) and 139% (*p* < 0.001), respectively. No significant differences in the relative risks of MI, unstable angina, and urgent revascularization were observed among the different age groups. Table 5 shows the cumulative incidence of MACEs at 12, 24, 36, and 48 months of follow-up by age group. The corresponding cumulative incidence plots for MACEs, MI, ischemic stroke, unstable angina, and urgent revascularization by age group after 48 months of follow-up are shown in Figure 2 and Figure 3.

## 4. Discussion

Over the past decades, the prevalence of T2DM has increased significantly and reached epidemic proportions [2,45]. People with T2DM, particularly older patients, are at a high risk of CVD, which is the leading cause of morbidity and mortality in these patients [13,25,46]. There is a wide body of evidence regarding age-related glucose dysregulation and CVD. Different processes driving metabolic dysregulation during aging have been described, such as insulin resistance, pancreatic β-cell impairment, and changes in physiological mechanisms, including the ability of insulin to suppress hepatic glucose output, peripheral glucose uptake, insulin pulsatility secretion, and response to incretins [47]. Moreover, insulin resistance, obesity, and other age-related factors such as dyslipidemia, inflammation, hypertension, autonomic dysfunction, and diminished vascular responsiveness have been described as contributors to CVD risk in T2DM [48]. However, elderly patients have often been excluded from or underrepresented in cardiovascular trials [26,27,28,29,30]. In this context, real-world data studies are important for quantifying the impact of age on the prevalence of cardiovascular disease in T2DM patients and for optimizing treatment management. This retrospective real-world study used NLP and ML to evaluate the cumulative incidence of MACE in patients with T2DM and stable CAD in Spain between 2014 and 2018. Our main results showed that in real life, 34.8% of this population is older than 75 years and is more likely to have comorbidities and receive pharmacological and interventional treatments. They are also more likely to have MACEs, especially ischemic stroke.

The median age of our population was 70.7 years, and approximately 35% of the patients were over the age of 75 years. Although there were 1.6 times more men than women in the total population, the male-to-female ratio narrowed to nearly 1:1 in older patients. Among other factors, the greater longevity of women results in their numerical predominance in older age groups, thereby narrowing the sex gap in older age. The median age and sex ratios in our study population were consistent with those reported in previously published studies [32,49,50]. Notably, our population included a higher proportion of patients aged >75 years than that in the US ATHENA study (35.0% vs. 25.0%) [50]. The representative inclusion of older men and women in this study may help to improve the detection of age-specific effects and assess the external validity of the study findings [26,27,28,29,30].

Patients with T2DM are more likely to have diffuse and multivessel vascular disease, a feature of advanced disease often associated with poor prognosis and outcomes [46,51,52]. These patients often require coronary revascularization in addition to pharmacological treatment [53]. In our study, the type of CAD was detected in more patients in the older age groups. As we have previously shown, half of the patients in this population underwent revascularization, and irrespective of age, almost three times as many patients underwent PCI as CABG [31]. Moreover, both PCI and CABG were more frequently observed in the older groups, with more comorbidities than in the younger groups. Clinical trials have shown that CABG yields better outcomes than PCI for all-cause mortality, cardiac mortality, and repeated revascularization in these patients [54,55,56,57], particularly in diabetic patients on insulin [58]. However, CABG is much more invasive than PCI and evidence in high-risk surgical patients is limited. This may partly explain the lower proportion of patients who underwent CABG in the present study.

Patients with T2DM and stable CAD but without prior stroke or MI have a high prevalence of comorbidities that significantly increase the likelihood of developing MACE [31,59]. Our results confirmed previous findings that the burden of comorbidity increases with age [60,61]. In addition to major comorbidities such as hypertension, angina, valvular heart disease, and peripheral vascular disease, we found that less common comorbidities in the general population were significantly increased in patients over 75 years of age compared to those under 65 years of age. These included atrial flutter, atrial fibrillation and heart failure. Patients with multiple comorbidities have reduced health-related quality of life [62,63] and require high levels of specific healthcare [64]. The prognosis and clinical outcomes of T2DM-CAD patients vary with individual variability and age, and diabetes care may remain suboptimal in many types of patients. It is therefore important to develop specific management strategies for common multimorbidity clusters that will help refine clinical decisions and enable patient-centered prevention and management [65], as recently recommended by the American Heart Association (AHA) [66].

In general, the pharmacological treatments prescribed for the management of T2DM and its associated comorbidities in our study population followed guideline-based recommendations. Regarding treatments for glycemic control, the use of metformin, the most commonly prescribed drug in all age groups, and insulin, decreased progressively with increasing age, whereas the use of sulfonylureas and DPP4 inhibitors increased. To achieve the appropriate therapeutic goal of maximizing optimal glycemic control while minimizing the risk of hypoglycemia, antidiabetic agents should be selected based on a comprehensive assessment of the circumstances of elderly patients [67,68]. Aspirin alone was the most prescribed antiplatelet therapy, and its use increased significantly with age. The use of other treatments, including statins to control hyperlipidemia and ACE inhibitors/ARBs or beta blockers for the management of arterial hypertension, also increased with increasing age, most likely reflecting the increasing occurrence of multimorbidity in older patients [69].

The presence of multimorbidity in patients with T2DM and stable CAD significantly increases the risk of developing MACEs [8,9,10,11]. In a previous study, we showed that, in this population, multivessel CAD, single-vessel CAD, PCI, transient ischemic attack, and heart failure were independently associated with the development of new MACEs [31]. Here, we showed that after 48 months of follow-up, patients over 75 years of age were 1.29 times significantly more likely to have a new MACE and, in particular, 2.39 times more likely to have an ischemic stroke than patients under 65 years of age. In this regard, the lack of significant differences in the relative risks of MI, unstable angina, or urgent revascularization between the different age groups in our study could be related to the difficulty in diagnosing patients with CAD, especially elderly individuals with diabetes [70]. Moreover, patients with T2DM may have different phenotypes of chronic coronary syndrome with different outcomes, including those without significant CAD and microvascular dysfunction [71]. Compared to the patients in our study with those randomized to the control arm in the phase 3 THEMIS trial (NCT01991795) or the PEGASUS-TIMI 54 trial (NCT01225562) [72,73], the age of our population was higher than that in the THEMIS (median age, 70.7 vs. 66.0 years) and PEGASUS-TIMI 54 (mean age, 70 ± 11.3 vs. 65.4 ± 8.3) trials, which may have contributed to the increased risk of cardiovascular outcomes in our population. The 36-month cumulative incidence of MACEs was remarkably higher in our study than in the THEMIS trial, being 2.8-fold higher for new MACEs (21.2% vs. 7.6%), 3.3-fold higher for MI (10.8% vs. 3.3%), and 2.8-fold higher for ischemic stroke (5.1% vs. 1.8%). It was also higher than that in the PEGASUS-TIMI 54 trial, being 2.3-fold higher for new MACE (21.2% vs. 9.0%), 2.0-fold higher for MI (10.8% vs. 5.3%), and 3.0-fold higher for ischemic stroke (5.1% vs. 1.7%). Table 6 shows patient demographic and clinical characteristics at baseline in the THEMIS and PEGASUS-TIMI 54 trials, as well as in the ACORDE study. A recently published study evaluated the prevalence of T2DM-CAD without prior MI or stroke and the risk of major outcomes in a real-world setting in France, specifically in a population that met selection criteria similar to those of the randomized THEMIS trial and close to those that would be applied in current practice for therapeutic indication after such a trial. When comparing our population with that of the French study, although both were similar in age, our patients had a higher prevalence of cardiovascular comorbidities, particularly heart failure, as well as other comorbidities, such as renal impairment and dyslipidemia. In addition, significantly more patients in our population were receiving anticoagulant and antiplatelet therapy, and more patients underwent PCI. In general, the 24-month cumulative incidences of major cardiovascular outcomes in our study were higher than those reported in the French study, although no formal comparison was made. This was particularly striking for the cumulative incidence of MI, which was five to six times higher both in the general population (7.8 vs. 1.3) and in the different age groups (patients <65 years: 7.9 vs. 1.3, patients 65–75 years: 7.6 vs. 1.2, patients >75 years: 7.9 vs. 1.5) [32]. Taken together, these data support previous evidence that T2DM-CAD patients enrolled in clinical trials may be younger and have a lower cardiovascular risk profile than those found in routine clinical practice, which may, to some extent, compromise the external validity of trial results and the actual treatment performance in real life.

Our study has several strengths. It is a multicenter study evaluating real-world data from patients with T2DM and CAD, a highly prevalent condition associated with the development of clinical complications leading to significant morbidity and mortality, particularly in the elderly. This study provides reliable data on the age-related cardiovascular risk profile of T2DM-CAD patients seen in routine clinical practice, as well as their management and clinical outcomes. In addition, our results further demonstrate that the findings of randomized clinical trials may not be fully applicable to all patients with diabetes, especially in the context of an aging population with increasing life expectancy. Moreover, by using NLP and ML, we were able to create an enriched dataset by extracting important information from unstructured free-text, not only limited to structured data such as those obtained using the International Classification of Diseases (ICD) coding as previously reported in real-life studies. This study also has limitations, which have been described in a previous article stemming from this study. Briefly, the findings of our study were limited by the availability and accuracy of the information on EHRs. Data extraction methods may also be intrinsically subject to reporting and information biases. Due to the retrospective nature of the study, based on real-world data, some potentially interesting variables were not properly documented and, therefore, not analyzed. Finally, although known confounders were considered, unknown confounders may have influenced the results of this study [31].

## 5. Conclusions

The disease burden in patients with T2DM and stable CAD increases with age. Older patients have more cardiovascular comorbidities, which increases the risk of major cardiovascular events. However, elderly patients have often been excluded from cardiovascular trials, including those that influence the current clinical guidelines. By analyzing readily available information in EHRs using NLP and ML technologies, we were able to provide real-world data on age-related characteristics and MACE in patients with T2DM and stable CAD in Spain. Our results suggest that T2DM-CAD patients in routine clinical practice tend to be older, have higher multimorbidity, are more treated, and have a higher risk of developing major cardiovascular outcomes than is commonly assumed from clinical trial data.

## Figures and Tables

**Figure 1 jcm-12-05218-f001:**
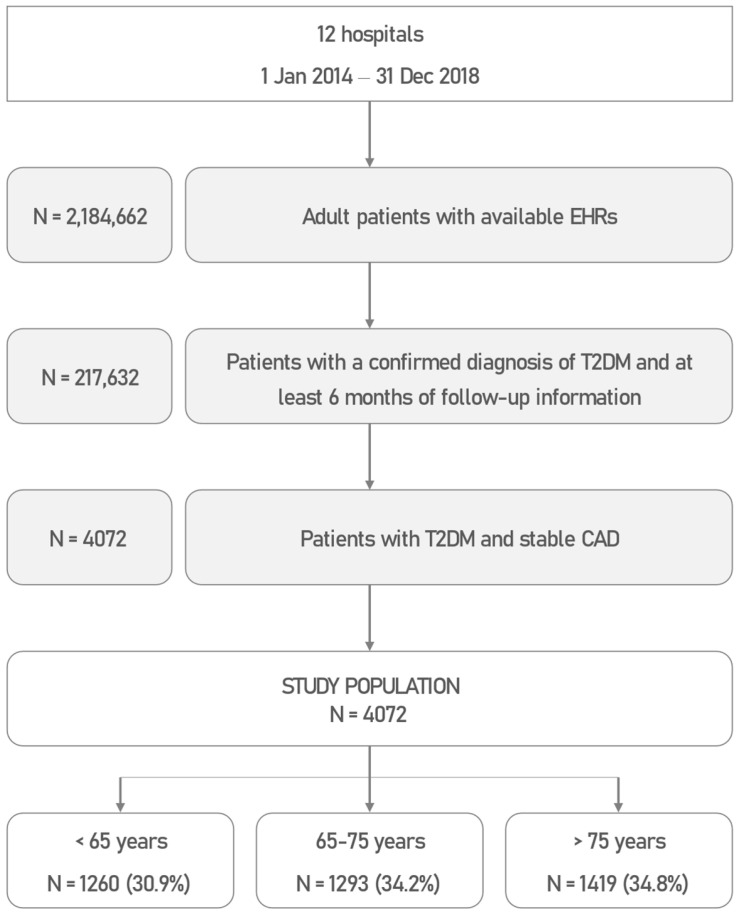
Patient Flowchart. Abbreviations: CAD: coronary artery disease; EHR: electronic health records; T2DM: type 2 diabetes mellitus.

**Figure 2 jcm-12-05218-f002:**
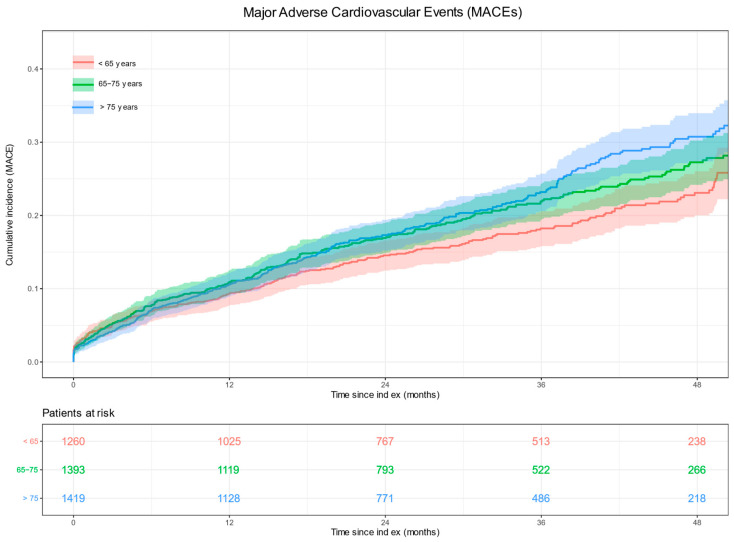
Cumulative incidence plot for major adverse cardiovascular events (MACEs) after 48 months of follow-up. The number of patients at risk across the follow-up period is indicated below.

**Figure 3 jcm-12-05218-f003:**
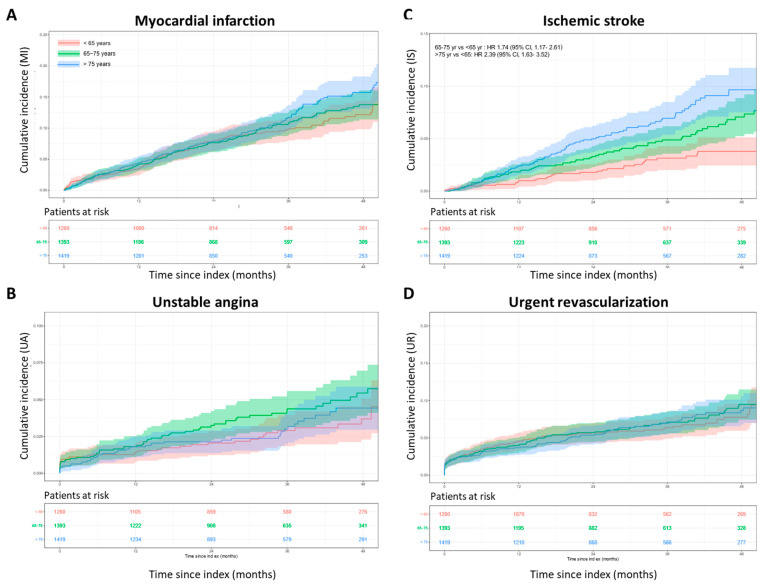
Cumulative incidence plot for myocardial infarction (**A**), ischemic stroke (**B**), unstable angina (**C**), and urgent revascularization (**D**) after 48 months of follow-up. The number of patients at risk across the follow-up period is indicated below.

**Table 1 jcm-12-05218-t001:** Patient demographic and clinical characteristics at baseline.

	All (*n* = 4072)	<65 yr (*n* = 1260)	65–75 yr (*n* = 1393)	>75 yr (*n* = 1419)	65–75 vs. <65	>75 vs. <65	>75 vs. 65–75
					OR ‡ (95% CI); *p*-Value		
Demographic Characteristics							
Male sex, *n* (%)	2531 (62.2)	835 (66.3)	925 (66.4)	771 (54.3)	1.01 (0.86, 1.18)	0.61 (0.52, 0.71)	0.60 (0.52, 0.70)
					*p* = 0.942	*p* < 0.001 *	*p* < 0.001 *
Smoking history, *n* (%)	2208 (54.2)	835 (66.3)	770 (55.3)	603 (42.5)	0.63 (0.54, 0.74)	0.38 (0.32, 0.44)	0.60 (0.51, 0.69)
					*p* < 0.001 *	*p* < 0.001 *	*p* < 0.001 *
Current smoker	713 (17.5)	332 (26.3)	226 (16.2)	155 (10.9)	0.54 (0.45, 0.65)	0.34 (0.28, 0.42)	0.63 (0.51, 0.79)
					*p* < 0.001 *	*p* < 0.001 *	*p* < 0.001 *
Former smoker	1495 (36.7)	503 (39.9)	544 (39.1)	448 (31.6)	0.96 (0.83, 1.13)	0.69 (0.59, 0.81)	0.72 (0.62, 0.84)
					*p* = 0.648	*p* < 0.001 *	*p* < 0.001 *
Never smoker/unknown	1864 (45.8)	425 (33.7)	623 (44.7)	816 (57.5)	1.59 (1.36, 1.86)	2.66 (2.27, 3.11)	1.67 (1.44, 1.94)
					*p* < 0.001 *	*p* < 0.001 *	*p* < 0.001 *
Clinical parameters							
BMI, kg/m^2^							
*n* (%)	696 (17.1)	309 (24.5)	254 (18.2)	133 (9.4)			
Median (Q1, Q3)	30.5 (26.9, 35.4)	31.6 (27.4, 37)	30.3 (26.9, 35.4)	29.3 (26.3, 32)	−1.01 (−2.28, 0.26) †	−2.17 (−3.73, −0.61) †	−1.16 (−2.77, 0.45) †
					*p* = 0.119	*p* = 0.006 *	*p* = 0.158
Type of CAD, n (%)							
Single-vessel CAD	813 (20.0)	216 (17.1)	276 (19.8)	321 (22.6)	1.19 (0.98, 1.46)	1.41 (1.17, 1.71)	1.18 (0.99, 1.42)
					*p* = 0.077	*p* < 0.001 *	*p* = 0.069
Multivessel CAD	1602 (39.3)	444 (35.2)	561 (40.3)	597 (42.1)	1.24 (1.06, 1.45)	1.33 (1.14, 1.56)	1.08 (0.93, 1.25)
					*p* = 0.008 *	*p* < 0.001 *	*p* = 0.332
Left main CAD	27 (0.7)	9 (0.7)	7 (0.5)	11 (0.8)	0.70 (0.25, 1.89)	1.09 (0.45, 2.70)	1.55 (0.61, 4.21)
					*p* = 0.484	*p* = 0.855	*p* = 0.368
Other/Unknown	1630 (40.0)	591 (46.9)	549 (39.4)	490 (34.5)	0.74 (0.63, 0.86)	0.60 (0.51, 0.70)	0.81 (0.70, 0.95)
					*p* < 0.001 *	*p* < 0.001 *	*p* = 0.007 *
LVEF, %							
*n* (%)	365 (9.0)	106 (8.4)	126 (9)	133 (9.4)			
Median (Q1, Q3)	51 (40, 62)	50 (35, 60)	55.5 (40, 65)	50 (40, 60)	4.37 (0.38, 8.36) †	3.23 (−0.71, 7.17) †	−1.14 (−4.90, 2.62) †
					*p* = 0.032 *⁋	*p* = 0.108	*p* = 0.552

Abbreviations: BMI: body mass index; CAD: coronary artery disease; 95% CI: confidence interval at the 95% confidence level; LVEF: left ventricular ejection fraction; OR: odds ratio; (Q1, Q3): interquartile range; SD: standard deviation; yr: years. ‡ odds ratio from a logistic regression; † coefficient from a linear regression; * differences were considered statistically significant when *p* < 0.05; ⁋ differences were not significant when adjusting *p*-values for multiple comparisons (Benjamini–Hochberg method).

**Table 2 jcm-12-05218-t002:** Patient analytical parameters and comorbidities at baseline.

	All (*n* = 4072)	<65 yr (*n* = 1260)	65–75 yr (*n* = 1393)	>75 yr (*n* = 1419)	65–75 vs. <65	>75 vs. <65	>75 vs. 65–75
					OR ‡ (95% CI); *p*-Value		
Analytical parameters							
Glucose, mg/dL							
*n* (%)	2749 (67.5)	888 (70.5)	933 (67)	928 (65.4)			
Median (Q1, Q3)	135 (113, 168)	135 (112, 175)	134 (114, 166)	134.5 (112, 164)	−3.44 (−9.08, 2.20) †	−4.82 (−10.46, 0.83) †	−1.38 (−6.96, 4.20) †
					*p* = 0.232	*p* = 0.095	*p* = 0.628
HbA1c, %							
*n* (%)	1987 (48.8)	694 (55.1)	677 (48.6)	616 (43.4)			
Median (Q1, Q3)	6.9 (6.3, 7.9)	7.0 (6.3, 8.1)	6.9 (6.3, 7.8)	6.9 (6.3, 7.7)	−0.23 (−0.39, −0.08) †	−0.20 (−0.36, −0.05) †	0.03 (−0.13, 0.19) †
					*p* = 0.003*	*p* = 0.011*	*p* = 0.731
Total cholesterol, mg/dL							
*n* (%)	1943 (47.7)	687 (54.5)	643 (46.2)	613 (43.2)			
Median (Q1, Q3)	160 (133, 193)	172 (141, 204)	156 (133, 188)	151 (126, 182)	−13.11 (−18.11, −8.10) †	−19.49 (−24.55, −14.42) †	−6.38 (−11.52, −1.23) †
					*p* < 0.001 *	*p* < 0.001 *	*p* = 0.015 *
HDL, mg/dL							
*n* (%)	1958 (48.1)	699 (55.5)	647 (46.4)	612 (43.1)			
Median (Q1, Q3)	42 (35, 51)	42 (35, 50)	42 (35, 50)	43 (36, 52)	−0.92 (−2.66, 0.82) †	0.04 (−1.73, 1.81) †	0.96 (−0.84, 2.76) †
					*p* = 0.299	*p* = 0.965	*p* = 0.294
LDL, mg/dL							
*n* (%)	1999 (49.1)	673 (53.4)	674 (48.4)	652 (45.9)			
Median (Q1, Q3)	85 (68, 110)	92 (71, 120)	84 (67, 107)	81 (65, 103)	−9.22 (−13.18, −5.26) †	−11.29 (−15.28, −7.29) †	−2.07 (−6.06, 1.93) †
					*p* < 0.001 *	*p* < 0.001 *	*p* = 0.311
Triglycerides, mg/dL							
*n* (%)	2073 (50.9)	733 (58.2)	691 (49.6)	649 (45.7)			
Median (Q1, Q3)	142 (99, 198)	155 (106, 216)	148 (102, 197)	121 (88, 175)	1.23 (−69.26, 71.72) †	10.78 (−60.87, 82.44) †	9.55 (−63.11, 82.22) †
					*p* = 0.973	*p* = 0.768	*p* = 0.797
Comorbidities, *n* (%)							
Arterial hypertension	3447 (84.7)	971 (77.1)	1191 (85.5)	1285 (90.6)	1.75 (1.44, 2.14)	2.85 (2.29, 3.57)	1.63 (1.29, 2.05)
					*p* < 0.001 *	*p* < 0.001 *	*p* < 0.001 *
Angina	1646 (40.4)	420 (33.3)	587 (42.1)	639 (45.0)	1.46 (1.24, 1.71)	1.64 (1.40, 1.92)	1.12 (0.97, 1.31)
					*p* < 0.001 *	*p* < 0.001 *	*p* = 0.122
Heart valve disease	1568 (38.5)	381 (30.2)	505 (36.3)	682 (48.1)	1.31 (1.12, 1.54)	2.13 (1.82, 2.50)	1.63 (1.40, 1.89)
					*p* = 0.001 *	*p* < 0.001 *	*p* < 0.001 *
Peripheral vascular disease	1513 (37.2)	339 (26.9)	511 (36.7)	663 (46.7)	1.57 (1.33, 1.86)	2.38 (2.03, 2.80)	1.51 (1.30, 1.76)
					*p* < 0.001 *	*p* < 0.001 *	*p* < 0.001 *
Heart failure	936 (23.0)	249 (19.8)	267 (19.2)	420 (29.6)	0.96 (0.79, 1.17)	1.71 (1.43, 2.04)	1.77 (1.49, 2.11)
					*p* = 0.699	*p* < 0.001 *	*p* < 0.001 *
Atrial fibrillation	590 (14.5)	86 (6.8)	192 (13.8)	312 (22.0)	2.18 (1.68, 2.86)	3.85 (3.00, 4.98)	1.76 (1.45, 2.15)
					*p* < 0.001 *	*p* < 0.001 *	*p* < 0.001 *
Peripheral artery disease	528 (13.0)	141 (11.2)	197 (14.1)	190 (13.4)	1.31 (1.04, 1.65)	1.23 (0.97, 1.55)	0.94 (0.76, 1.16)
					*p* = 0.023*	*p* = 0.085	*p* = 0.563
Hyperlipidemia	1666 (40.9)	553 (43.9)	599 (43.0)	514 (36.2)	0.96 (0.83, 1.12)	0.73 (0.62, 0.85)	0.75 (0.65, 0.88)
					*p* = 0.645	*p* < 0.001 *	*p* < 0.001 *
Obesity	1328 (32.6)	545 (43.3)	465 (33.4)	318 (22.4)	0.66 (0.56, 0.77)	0.38 (0.32, 0.45)	0.58 (0.49, 0.68)
					*p* < 0.001 *	*p* < 0.001 *	*p* < 0.001 *
Anemia	867 (21.3)	193 (15.3)	285 (20.5)	389 (27.4)	1.42 (1.16, 1.74)	2.09 (1.72, 2.54)	1.47 (1.23, 1.75)
					*p* = 0.001 *	*p* < 0.001 *	*p* < 0.001 *
Chronic kidney disease	740 (18.2)	145 (11.5)	235 (16.9)	360 (25.4)	0.53 (0.32, 0.87)	0.44 (0.25, 0.73)	0.82 (0.45, 1.47)
					*p* = 0.013 *	*p* = 0.002 *	*p* = 0.511
Depression/anxiety	820 (20.1)	306 (24.3)	258 (18.5)	256 (18.0)	0.71 (0.59, 0.85)	0.69 (0.57, 0.83)	0.97 (0.80, 1.17)
					*p* < 0.001 *	*p* < 0.001 *	*p* = 0.742
COPD/asthma	699 (17.2)	195 (15.5)	253 (18.2)	251 (17.7)	1.21 (0.99, 1.49)	1.17 (0.96, 1.44)	0.97 (0.80, 1.17)
					*p* = 0.065	*p* = 0.125	*p* = 0.743

Abbreviations: 95% CI: confidence interval at the 95% confidence level; COPD: chronic obstructive pulmonary disease; HbA1c: glycated hemoglobin, type A1c; HDL: high-density lipoprotein cholesterol; LDL: low-density lipoprotein cholesterol; OR: odds ratio; (Q1, Q3): interquartile range; SD: standard deviation; yr: years. ‡ odds ratio from a logistic regression; † coefficient from a linear regression; * differences were considered statistically significant when *p* < 0.05.

**Table 3 jcm-12-05218-t003:** Pharmacological treatments.

	All (*n* = 4072)	<65 yr (*n* = 1260)	65–75 yr (*n* = 1393)	>75 yr (*n* = 1419)	65–75 vs. <65	>75 vs. < 65	>75 vs. 65–75
					OR ‡ (95% CI); *p*-Value		
Insulin treatment	1018 (25.0)	354 (28.1)	341 (24.5)	323 (22.8)	0.83 (0.70, 0.99)	0.75 (0.63, 0.90)	0.91 (0.76, 1.08)
					*p* = 0.035 *⁋	*p* = 0.002 *	*p* = 0.284
LA insulin	795 (19.5)	285 (22.6)	257 (18.4)	253 (17.8)	0.77 (0.64, 0.93)	0.74 (0.61, 0.90)	0.96 (0.79, 1.16)
					*p* = 0.008 *	*p* = 0.002 *	*p* = 0.670
FA insulin	345 (8.5)	155 (12.3)	110 (7.9)	80 (5.6)	0.61 (0.47, 0.79)	0.43 (0.32, 0.56)	0.70 (0.52, 0.94)
					*p* < 0.001 *	*p* < 0.001 *	*p* = 0.018 *
Intermediate or LA insulin + FA insulin	219 (5.4)	68 (5.4)	75 (5.4)	76 (5.4)	>0.99 (0.71, 1.40)	0.99 (0.71, 1.39)	0.99 (0.72, 1.38)
					*p* = 0.988	*p* = 0.963	*p* = 0.974
Intermediate-acting insulin	104 (2.6)	36 (2.9)	40 (2.9)	28 (2.0)	1.01 (0.64, 1.59)	0.68 (0.41, 1.13)	0.68 (0.41, 1.11)
					*p* = 0.982	*p* = 0.137	*p* = 0.123
Oral hypoglycemic agents	4072 (100.0)	1260 (100.0)	1393 (100.0)	1419 (100.0)	**	**	**
Metformin	3160 (77.6)	1012 (80.3)	1086 (78.0)	1062 (74.8)	0.87 (0.72, 1.05)	0.73 (0.61, 0.88)	0.84 (0.71, <1.01)
					*p* = 0.136	*p* = 0.001 *	*p* = 0.052
Sulfonylureas	881 (21.6)	223 (17.7)	327 (23.5)	331 (23.3)	1.43 (1.18, 1.73)	1.41 (1.17, 1.71)	0.99 (0.83, 1.18)
					*p* < 0.001 *	*p* < 0.001 *	*p* = 0.926
DPP4i	848 (20.8)	235 (18.7)	283 (20.3)	330 (23.3)	1.11 (0.92, 1.35)	1.32 (1.10, 1.60)	1.19 (0.99, 1.42)
					*p* = 0.280	*p* = 0.004	*p* = 0.059
Glinidines	507 (12.5)	141 (11.2)	162 (11.6)	204 (14.4)	1.04 (0.82, 1.33)	1.33 (1.06, 1.68)	1.28 (1.02, 1.59)
					*p* = 1.723	*p* = 0.014 *	*p* = 0.031 *
GLP1-RA	212 (5.2)	128 (10.2)	66 (4.7)	18 (1.3)	0.44 (0.32, 0.60)	0.11 (0.07, 0.18)	0.26 (0.15, 0.43)
					*p* < 0.001 *	*p* < 0.001 *	*p* < 0.001 *
iSGLT2	106 (2.6)	58 (4.6)	32 (2.3)	16 (1.1)	0.49 (0.31, 0.75)	0.24 (0.13, 0.40)	0.49 (0.26, 0.87)
					*p* = 0.001 *	*p* < 0.001 *	*p* = 0.019 *
Thiazolidinediones	91 (2.2)	27 (2.1)	36 (2.6)	28 (2)	1.21 (0.73, 2.02)	0.92 (0.54, 1.57)	0.76 (0.46, 1.25)
					*p* = 0.456	*p* = 0.757	*p* = 0.279
Alpha glucosidase	57 (1.4)	9 (0.7)	22 (1.6)	26 (1.8)	2.23 (1.06, 5.12)	2.59 (1.26, 5.88)	1.16 (0.66, 2.08)
					*p* = 0.044 *	*p* = 0.014 *	*p* = 0.605
Anticoagulant therapy	776 (19.1)	165 (13.1)	252 (18.1)	359 (25.3)	1.47 (1.19, 1.82)	2.25 (1.84, 2.76)	1.53 (1.28, 1.84)
					*p* < 0.001 *	*p* < 0.001 *	*p* < 0.001 *
Antiplatelet agents	2843 (69.8)	778 (61.7)	1008 (72.4)	1057 (74.5)	1.62 (1.38, 1.91)	1.81 (1.53, 2.13)	1.12 (0.94, 1.32)
					*p* < 0.001 *	*p* < 0.001 *	*p* = 0.202
ASA	2606 (64.0)	737 (58.5)	923 (66.3)	946 (66.7)	1.39 (1.19, 1.63)	1.42 (1.21, 1.66)	1.02 (0.87, 1.19)
					*p* < 0.001 *	*p* < 0.001 *	*p* = 0.819
Clopidogrel	1208 (29.7)	324 (25.7)	420 (30.2)	464 (32.7)	1.25 (1.05, 1.48)	1.40 (1.19, 1.66)	1.13 (0.96, 1.32)
					*p* = 0.011 *	*p* < 0.001 *	*p* = 0.146
Dual antiplatelet therapy	1027 (25.2)	300 (23.8)	362 (26)	365 (25.7)	1.12 (0.94, 1.34)	1.11 (0.93, 1.32)	0.99 (0.83, 1.17)
					*p* = 0.196	*p* = 0.253	*p* = 0.873
Clopidogrel + ASA	830 (20.4)	235 (18.7)	285 (20.5)	310 (21.8)	1.12 (0.93, 1.36)	1.22 (1.01, 1.47)	1.09 (0.91, 1.30)
					*p* = 0.241	*p* = 0.040 *⁋	*p* = 0.368
Other cardiovascular therapy	3922 (96.3)	1173 (93.1)	1351 (97.0)	1398 (98.5)	2.39 (1.65, 3.51)	4.94 (3.11, 8.21)	2.07 (1.23, 3.58)
					*p* < 0.001 *	*p* < 0.001 *	*p* = 0.007 *
ACE inhibitors or ARB	3282 (80.6)	930 (73.8)	1123 (80.6)	1229 (86.6)	1.48 (1.23, 1.77)	2.30 (1.89, 2.80)	1.56 (1.27, 1.91)
					*p* < 0.001 *	*p* < 0.001 *	*p* < 0.001 *
ACE inhibitors	2153 (52.9)	637 (50.6)	707 (50.8)	809 (57.0)	1.01 (0.87, 1.17)	1.30 (1.11, 1.51)	1.29 (1.11, 1.49)
					*p* = 0.919	*p* = 0.001 *	*p* = 0.001 *
ARB	1895 (46.5)	475 (37.7)	674 (48.4)	746 (52.6)	1.55 (1.33, 1.81)	1.83 (1.57, 2.14)	1.18 (1.02, 1.37)
					*p* < 0.001 *	*p* < 0.001 *	*p* = 0.026 *⁋
Beta blockers	2418 (59.4)	713 (56.6)	819 (58.8)	886 (62.4)	1.09 (0.94, 1.28)	1.28 (1.09, 1.49)	1.17 (<1.01, 1.36)
					*p* = 0.251	*p* = 0.002 *	*p* = 0.048 *⁋
Calcium channel blockers	1686 (41.4)	401 (31.8)	586 (42.1)	699 (49.3)	1.56 (1.33, 1.82)	2.08 (1.78, 2.44)	1.34 (1.15, 1.55)
					*p* < 0.001 *	*p* < 0.001 *	*p* < 0.001 *
Nitrates	1327 (32.6)	319 (25.3)	440 (31.6)	568 (40.0)	1.36 (1.15, 1.61)	1.97 (1.67, 2.32)	1.45 (1.24, 1.69)
					*p* < 0.001 *	*p* < 0.001 *	*p* < 0.001 *
Diuretics	1914 (47.0)	469 (37.2)	618 (44.4)	827 (58.3)	1.34 (1.15, 1.57)	2.36 (2.02, 2.75)	1.75 (1.51, 2.03)
					*p* < 0.001 *	*p* < 0.001 *	*p* < 0.001 *
Lipid-lowering drugs	3386 (83.2)	1019 (80.9)	1188 (85.3)	1179 (83.1)	1.37 (1.12, 1.68)	1.16 (0.95, 1.42)	0.85 (0.69, 1.04)
					*p* = 0.002 *	*p* = 0.136	*p* = 0.111
Statins	3223 (79.2)	947 (75.2)	1140 (81.8)	1136 (80.1)	1.49 (1.24, 1.80)	1.33 (1.11, 1.59)	0.89 (0.74, 1.08)
					*p* < 0.001 *	*p* = 0.002 *	*p* = 0.229
Other lipid-lowering drugs	885 (21.7)	361 (28.7)	314 (22.5)	210 (14.8)	0.72 (0.61, 0.86)	0.43 (0.36, 0.52)	0.60 (0.49, 0.72)
					*p* < 0.001 *	*p* < 0.001 *	*p* < 0.001 *

Abbreviations: ACE: angiotensin-converting enzyme; GLP1-RA: glucagon-like peptide 1 receptor agonist; iSGLT2: sodium–glucose cotransporter 2 inhibitor; ASA: acetylsalicylic acid; ARB: angiotensin receptor blocker; 95% CI: confidence interval at the 95% confidence level; DPPi: dipeptidyl peptidase-4 inhibitors; FA: fast-acting; LA: long-acting; OR: odds ratio; yr: years. ‡ odds ratio from a logistic regression; * differences were considered statistically significant when *p* < 0.05; ⁋ differences were not significant when adjusting *p*-values for multiple comparisons (Benjamini–Hochberg method); ** no variability.

**Table 4 jcm-12-05218-t004:** Coronary interventional treatments.

	All (*n* = 4072)	<65 yr (*n* = 1260)	65–75 yr (*n* = 1393)	>75 yr (*n* = 1419)	65–75 vs. <65	>75 vs. <65	>75 vs. 65–75
					OR ‡ (95% CI); *p*-Value		
Revascularization treatment	2016 (49.5)	564 (44.8)	710 (51)	742 (52.3)	1.28 (1.10, 1.49)	1.35 (1.16, 1.58)	1.05 (0.91, 1.22)
					*p* = 0.001 *	*p* <0.001 *	*p* = 0.483
PCI	1579 (38.8)	458 (36.3)	558 (40.1)	563 (39.7)	1.17 (<1.01, 1.37)	1.15 (0.98, 1.35)	0.98 (0.85, 1.14)
					*p* = 0.050 *⁋	*p* = 0.077	*p* = 0.836
CABG	585 (14.4)	147 (11.7)	218 (15.6)	220 (15.5)	1.40 (1.12, 1.76)	1.39 (1.11, 1.74)	0.99 (0.81, 1.21)
					*p* = 0.003 *	*p* = 0.004 *	*p* = 0.915

Abbreviations: CABG: coronary artery bypass grafting; 95% CI: confidence interval at the 95% confidence level; OR: odds ratio; PCI: percutaneous coronary intervention; yr: years. ‡ odds ratio from a logistic regression; * differences were considered statistically significant when *p* < 0.05; ⁋ differences were not significant when adjusting *p*-values for multiple comparisons (Benjamini–Hochberg method).

**Table 5 jcm-12-05218-t005:** Cumulative incidence of major adverse cardiovascular events.

	12 Months			24 Months			36 Months			48 Months			65–75 vs. <65	>75 vs. <65	>75 vs. 65–75
	<65 yr	65–75 yr	>75 yr	<65 yr	65–75 yr	>75 yr	<65 yr	65–75 yr	>75 yr	<65 yr	65–75 yr	>75 yr	HR ‡ (95% CI); *p*-Value		
MACEs	9.34	10.85	10.55	14.55	16.93	17.32	18.23	21.91	23.19	23.08	27.26	30.75	1.18 (>0.99, 1.40)	1.29 (1.09, 1.52)	1.09 (0.93, 1.27)
													*p* = 0.051	*p* = 0.003 *	*p* = 0.301
Myocardial infarction	3.48	4.19	4.41	7.92	7.64	7.86	9.89	10.72	11.72	12.14	13.75	15.73	1.07 (0.84, 1.36)	1.20 (0.94, 1.51)	1.12 (0.89, 1.40)
													*p* = 0.585	*p* = 0.139	*p* = 0.334
Stroke	1.97	3.37	3.49	3.68	5.67	6.82	5.91	8.24	9.82	8.62	11.01	14.19	1.39 (1.04, 1.87)	1.77 (1.33, 2.35)	1.27 (0.99, 1.63)
													*p* = 0.027 *⁋	*p* < 0.001 *	*p* = 0.063
Ischemic stroke	0.99	1.83	2.41	1.79	3.29	5.00	3.14	4.89	6.95	3.79	7.08	9.67	1.74 (1.17, 2.61)	2.39 (1.63, 3.52)	1.37 (1.01, 1.87)
													*p* = 0.007 *	*p* < 0.001 *	*p* = 0.045 *⁋
Unstable angina	1.36	1.81	1.80	2.05	3.24	2.14	2.90	4.37	3.01	3.69	5.45	4.43	1.52 (>0.99, 2.31)	1.10 (0.70, 1.73)	0.73 (0.49, 1.08)
													*p* = 0.051	*p* = 0.672	*p* = 0.113
Urgent revascularization	4.50	3.98	3.69	5.66	5.70	5.35	6.38	7.10	6.98	7.76	9.49	8.67	1.04 (0.79, 1.39)	0.97 (0.73, 1.30)	0.93 (0.70, 1.23)
													*p* = 0.763	*p* = 0.838	*p* = 0.603

Abbreviations: 95% CI: confidence interval at the 95% confidence level; HR: hazard ratio; MACEs: major adverse cardiovascular events; yr: years. ‡ hazard ratio from Cox proportional hazards regression models after 48 months of follow-up; * differences were considered statistically significant when *p* < 0.05; ⁋ differences were not significant when adjusting *p* values for multiple comparisons (Benjamini–Hochberg method).

**Table 6 jcm-12-05218-t006:** Patient demographic and clinical characteristics at baseline in THEMIS and PEGASUS-TIMI 54 trials and ACORDE study.

	THEMIS Trial		PEGASUS-TIMI 54 Trial			ACORDE Study
	Placebo (*n* = 9601)	Ticagrelor (*n* = 9619)	Placebo (*n* = 7067)	Ticagrelor, 60 mg (*n* = 7045)	Ticagrelor, 90 mg (*n* = 7050)	All (*n* = 4072)
Age, years						
Mean (SD)	—	—	65.4 ± 8.4	65.2 ± 8.4	65.4 ± 8.3	70 ± 11.3
Median (IQR)	66.0 (61.0–72.0)	66.0 (61.0–72.0)	—	—	—	70.7 (62.9–78.1)
Male sex, *n* (%)	6613 (68.9)	6576 (68.4)	5385 (76.2)	5384 (76.4)	5333 (75.6)	2531 (62.2)
Median BMI, kg/m^2^ (IQR)	29.1 (26.0–32.8)	29.0 (26.1–32.6)	—	—	—	30.5 (26.9–35.4) ^‡^
Weight, Kg	—	—	82.0 ± 16.7	82.0 ± 17.0	81.8 ± 16.6	—
Current smoker, *n* (%)	1038 (10.8)	1056 (11.0)	1187 (16.8)	1206 (17.1)	1143 (16.2)	713 (17.5)
Comorbidities, *n* (%)						
Hypertension	8867 (92.4)	8909 (92.6)	5462 (77.5)	5461 (77.5)	5484 (77.6)	3447 (84.7)
Dyslipidemia	8367 (87.1)	8386 (87.2)	—	—	—	—
Hyperlipidemia	—	—	5410 (76.7)	5380 (76.4)	5451 (77.1)	1666 (40.9)
Cardiovascular events, *n* (incidence)						
MACEs	818 (8.5)	736 (7.7)	493 (7.85)	487 (7.77)	578 (9.04)	858 (21.1)
Cardiovascular death	357 (3.7)	364 (3.8)	182 (2.94)	174 (2.86)	210 (3.39)	—
Myocardial infarction	328 (3.4)	274 (2.8)	275 (4.40)	285 (4.53)	338 (5.25)	424 (10.4)
Ischemic stroke	191 (2.0)	152 (1.6)	88 (1.41)	78 (1.28)	103 (1.65)	198 (4.9)
Coronary arterial revascularization	879 (9.2)	828 (8.6)	74 (1.16)	62 (0.95)	76 (1.13)	2016 (49.5)

Abbreviations: BMI: body mass index; IQR: interquartile range; MACEs: major adverse cardiovascular events; SD: standard deviation; TIMI: thrombolysis in myocardial infarction. ^‡^ *n* = 696 patients (17.1%) with available BMI data. THEMIS trial, ClinicalTrials.gov number NCT01991795; PEGASUS-TIMI 54 trial, ClinicalTrials.gov number NCT01225562.

## Data Availability

Data are available on reasonable request to the authors. Thereafter, the committee of the project, together with the Ethics Committee of the hospitals involved, will assess the proposal and potentially proceed to the data sharing.

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
