# Peer review of "Impact of Advanced Age on the Incidence of Major Adverse Cardiovascular Events in Patients with Type 2 Diabetes Mellitus and Stable Coronary Artery Disease in a Real-World Setting in Spain"

_jcm, 2023, doi:10.3390/jcm12165218_

Round 1

Reviewer 1 Report

Here my comments related to manuscript from González-Juanatey et al.  titled “Impact of advanced age on the incidence of major ad verse cardiovascular events in patients with type 2 diabetes mellitus and stable coronary artery disease in a real-world setting in Spain”

-In general the manuscript is well written, but there are some phrases with grammatical errors and some typos.

-In the abstract section, the authors should add the time period of study and the percentage of sex. Why were the patients classified in these age groups?

-I recommend the next good reference (PMID: 34769060) for the introduction section of this interesting manuscript.

-Please add in the reference 4 the link that leads directly to the page.

-With respect to data related to mortality associated with type 2 diabetes mellitus, reference 5 is from the year 2013. It is possible to update these data.

-In the introduction section, it is necessary to add the most prevalent types of cardiovascular diseases reported in advanced age. For example, the authors could add results reported in the reference 26. Please also add some pathological processes activated by diabetes, which cause cardiovascular damage.  

-In the materials and methods, it is important to describe the diagnostic methods used to diagnose hypertension, angina, heart valve disease, and peripheral vascular disease. Please also include the time in years that patients were diagnosed with type 2 diabetes mellitus and coronary artery disease.

-In the following phrase (page 3, line 132-135): “The cumulative incidence of MACE, defined by the presence of MI, stroke, hospitalization for unstable angina, and urgent coronary revascularization [27], was analyzed during the follow-up age group.” Please add the period time of the study.

-In the table 2 information is missing (Oral hypoglycemic agents), and the table 3 is not complete.

-Figure 2 and 3 need better resolution.

-In discussion section, the authors should add information related to pathological processes, which are characteristic of aging and are accelerated by diabetes to cause cardiovascular disease.

English Language is fine, only some grammatical mistakes and some typos. 

Author Response

August 2nd, 2023

Dear Reviewer,

We would like to thank you for the opportunity to revise our manuscript entitled "Impact of advanced age on the incidence of major adverse cardiovascular events in patients with type 2 diabetes mellitus and stable coronary artery disease in a real-world setting in Spain".

Based on the comments from the Managing Editor and the three Reviewers, we have implemented some changes in the manuscript to further improve its content and structure. We address them below and provide a point-by-point answer.

We also provide a version of the manuscript using tracked changes and a clean version in word format.

We hope that our manuscript is now suitable for publication in JCM.

Yours sincerely, on behalf of all authors,

Dr González-Juanatey

Section Managing Editor:

In order to increase the visibility of your paper, could you please provide a graphical abstract (GA) in the form of a self-explanatory image to appear alongside the text abstract in the website?

Response: We have included a graphical abstract and thank the Section Manager Editor for this suggestion.

Reviewer 1:

Here my comments related to manuscript from González-Juanatey et al.  titled “Impact of advanced age on the incidence of major adverse cardiovascular events in patients with type 2 diabetes mellitus and stable coronary artery disease in a real-world setting in Spain”

-In general the manuscript is well written, but there are some phrases with grammatical errors and some typos.

Response: We thank Reviewer 1 for taking the time to review our work. We have reviewed the text and hope all grammatical errors have been corrected.

-In the abstract section, the authors should add the time period of study and the percentage of sex. Why were the patients classified in these age groups?

Response: We have added the study period and sex frequency in the abstract. We have selected these age groups for two main reasons. First, they divide the study population into three balanced groups. Second, it makes our results more easily comparable to other published studies (1). We have now included these method’s criteria in the new manuscript version (methods section).

(1) Blin P., Darmon P., Henry P., Guiard E., Bernard M.A., Dureau-Pournin C., Maizi H., Thomas-Delecourt F., Lassalle R., Droz-Perroteau C., et al. Patients with stable coronary artery disease and type 2 diabetes but without prior myocardial infarction or stroke and THEMIS-like patients: real-world prevalence and risk of major outcomes from the SNDS French nationwide claims database. Cardiovasc Diabetol. 2021, 20, 229

-I recommend the next good reference (PMID: 34769060) for the introduction section of this interesting manuscript.

Response: We have added this reference as suggested.

-Please add in the reference 4 the link that leads directly to the page.

Response: We have added the direct link as requested.

-With respect to data related to mortality associated with type 2 diabetes mellitus, reference 5 is from the year 2013. It is possible to update these data.

Response: We have updated reference 5 as suggested.

-In the introduction section, it is necessary to add the most prevalent types of cardiovascular diseases reported in advanced age. For example, the authors could add results reported in the reference 26. Please also add some pathological processes activated by diabetes, which cause cardiovascular damage.  

Response: We have added the information requested to the introduction.

-In the materials and methods, it is important to describe the diagnostic methods used to diagnose hypertension, angina, heart valve disease, and peripheral vascular disease. Please also include the time in years that patients were diagnosed with type 2 diabetes mellitus and coronary artery disease.

Response: We appreciate the Reviewer comment. All the included variables in the study, including comorbidities, were ascertained using NLP from unstructured free text information in patient’s EHRs. Therefore, the diagnosis of these characteristics was based on the reporting of the condition in the patient’s EHRs, which should be correlated with clinical criteria in most cases. We have included these details in the new version of the manuscript (“Extraction of the clinical data from EHR”).

Regarding the age of the study patients at the time of T2DM and CAD diagnosis, we have included these data in the Results section.

-In the following phrase (page 3, line 132-135): “The cumulative incidence of MACE, defined by the presence of MI, stroke, hospitalization for unstable angina, and urgent coronary revascularization [27], was analyzed during the follow-up age group.” Please add the period time of the study.

Response:  We have added study period information in the Study design section as requested.

-In the table 2 information is missing (Oral hypoglycemic agents), and the table 3 is not complete.

Response: We thank the Reviewer for the comment. We have added in Table 2 other oral hypoglycemic agents analyzed such as glinidines, GLP1-RA (Glucagon-Like Peptide 1 Receptor Agonist), iSGLT2 (Sodium-glucose co-transporter 2 inhibitor), Thiazolidinediones and Alpha-glucosidase.

Table 3 shows the cumulative incidence of major adverse cardiovascular events or MACE (12, 24, 36 and 48 months) including the corresponding HR (95% CI) between groups (65-75 vs <65; >75 vs <65 and >75 vs 65-75). We are sorry for the inconveniences, and we will confirm a correct table uploading in the next submission.

-Figure 2 and 3 need better resolution.

Response:  We thank the Reviewer for pointing this out, the image was corrupted during the manuscript submission, we hope this has been corrected. For the final version, figures will be submitted in high quality and individual files.

-In discussion section, the authors should add information related to pathological processes, which are characteristic of aging and are accelerated by diabetes to cause cardiovascular disease.

Response: We have added the information requested to the discussion.

Reviewer 2 Report

Impact of advanced age on the incidence of major adverse cardiovascular events in patients with type 2 diabetes mellitus and stable coronary artery disease in a real-world setting in Spain. It is merit, however, the following should be paid attention to:

-Due to too many abbreviations, abbreviations should be listed before the References section.

Author Response

August 2nd, 2023

Dear Reviewer,

We would like to thank you for the opportunity to revise our manuscript entitled "Impact of advanced age on the incidence of major adverse cardiovascular events in patients with type 2 diabetes mellitus and stable coronary artery disease in a real-world setting in Spain".

Based on the comments from the Managing Editor and the three Reviewers, we have implemented some changes in the manuscript to further improve its content and structure. We address them below and provide a point-by-point answer.

We also provide a version of the manuscript using tracked changes and a clean version in word format.

We hope that our manuscript is now suitable for publication in JCM.

Yours sincerely, on behalf of all authors,

Dr González-Juanatey

Section Managing Editor:

In order to increase the visibility of your paper, could you please provide a graphical abstract (GA) in the form of a self-explanatory image to appear alongside the text abstract in the website?

Response: We have included a graphical abstract and thank the Section Manager Editor for this suggestion.

Reviewer 2:

Impact of advanced age on the incidence of major adverse cardiovascular events in patients with type 2 diabetes mellitus and stable coronary artery disease in a real-world setting in Spain. It is merit, however, the following should be paid attention to:

-Due to too many abbreviations, abbreviations should be listed before the References section.

Response:  We thank Reviewer 2 for these comments. We have added an abbreviation list to the manuscript.

Reviewer 3 Report

In this study, researchers aimed to provide real-world data on patients with type 2 diabetes mellitus and coronary artery disease in Spain. They used EHRead® technology, based on natural language processing and machine learning, to extract information from electronic health records of 4,072 patients from 12 hospitals between 2014 and 2018. They showed that older patients (above 75) had more comorbidities, received extensive treatment, and faced a higher risk of major cardiovascular events and ischemic stroke compared to younger patients. This highlights the importance of age-specific management for better outcomes. 

I really appreciate the comparison between the THEMIS and PEGASUS-TIMI 54 trials and the ACORDE study. However, the major findings that MACE and comorbidities are correlated with age in patients with T2DM and CAD are not a true novelty in the scientific panorama

The strengths of the study are the sample size and the fact that real word data were used.

However, some major issues need to be addressed:

Introduction (Line 95-105). Please consider also that new data regarding the cardioprotective role of new SLGT2 inhibitors in patients with CAD / acute myocardial infarction are available (10.1016/j.diabres.2023.110766; 10.1016/j.phrs.2022.106597). Please consider expanding this introduction section. 

 “Although included in this definition, all-cause or cardiovascular death was not analyzed because the nature of the data source did not allow their occurrence to be accurately identified”. This should be explained better and included in the limitations section. In fact, I don’t understand why the AA decided to exclude all-cause death in the MACE calculation. I know that is difficult to recognise real cardiovascular death, but at least all-cause death must be included in the MACE definition. 

Also: Why the AA excluded patients with anticoagulant treatment? Why they performed a cross-sectional analysis stratified according to age (<65 years, 65-75 years and >75 years)? Please explain in detail these methods' criteria

Results and Discussion:

Table 1 is quite difficult to read. I suggest splitting this table. The first one only includes the three groups with a single p-value and a Bonferroni correction. The second table for the OR and linear regression. The same for Table 2.

Explain this finding in the discussion section: “No significant differences in the relative risks of MI, unstable angina and urgent revascularisation were observed between the different age groups.” I suggest that the diagnosis and treatment of CAD are difficult, especially in patients with T2DM and older age (https://doi.org/10.1016/j.jacc.2019.07.083; 10.1016/j.ijcard.2022.07.038; http://dx.doi.org/10.1016/j.jcmg.2016.01.039; http://dx.doi.org/10.1016/j.jcmg.2014.08.001; 10.1093/ehjci/jead046). Moreover, patients with T2DM could have different phenotypes of chronic coronary syndrome with different outcomes, including those without significant CAD and microvascular dysfunction (10.1016/j.jcmg.2022.12.029; 10.2459/JCM.0000000000001305; https://doi.org/10.1016/j.jacc.2021.07.042).

It could be useful to perform a multivariable COX analysis including age, different treatments (focus on Aspirin) and comorbidities to search independent predictors of MACE. Furthermore, a comparison in terms of MACE between T2DM CAD and NoCAD patients could be very interesting.

Please improve the Figures. Why there is a different number at risk at time 0 between figures 2 and 3?

Minor suggestion: 

Please correct line 103: “and references 103 therein).”

I suggest improving the fluidity and clarity of the discussion. In addition, a double check of English and abbreviations is suggested.

Author Response

August 2nd, 2023

Dear Reviewer,

We would like to thank you for the opportunity to revise our manuscript entitled "Impact of advanced age on the incidence of major adverse cardiovascular events in patients with type 2 diabetes mellitus and stable coronary artery disease in a real-world setting in Spain".

Based on the comments from the Managing Editor and the three Reviewers, we have implemented some changes in the manuscript to further improve its content and structure. We address them below and provide a point-by-point answer.

We also provide a version of the manuscript using tracked changes and a clean version in word format.

We hope that our manuscript is now suitable for publication in JCM.

Yours sincerely, on behalf of all authors,

Dr González-Juanatey

Section Managing Editor:

In order to increase the visibility of your paper, could you please provide a graphical abstract (GA) in the form of a self-explanatory image to appear alongside the text abstract in the website?

Response: We have included a graphical abstract and thank the Section Manager Editor for this suggestion.

Reviewer 3:

In this study, researchers aimed to provide real-world data on patients with type 2 diabetes mellitus and coronary artery disease in Spain. They used EHRead® technology, based on natural language processing and machine learning, to extract information from electronic health records of 4,072 patients from 12 hospitals between 2014 and 2018. They showed that older patients (above 75) had more comorbidities, received extensive treatment, and faced a higher risk of major cardiovascular events and ischemic stroke compared to younger patients. This highlights the importance of age-specific management for better outcomes. 

I really appreciate the comparison between the THEMIS and PEGASUS-TIMI 54 trials and the ACORDE study. However, the major findings that MACE and comorbidities are correlated with age in patients with T2DM and CAD are not a true novelty in the scientific panorama

The strengths of the study are the sample size and the fact that real word data were used.

Response:  We appreciate the words from Reviewer 3 and their time spent reviewing our work.

However, some major issues need to be addressed:

Introduction (Line 95-105). Please consider also that new data regarding the cardioprotective role of new SLGT2 inhibitors in patients with CAD / acute myocardial infarction are available (10.1016/j.diabres.2023.110766; 10.1016/j.phrs.2022.106597). Please consider expanding this introduction section. 

 Response:  We thank the Reviewer for this comment. We have expanded the Introduction section with this concept and the references provided, as requested.

 “Although included in this definition, all-cause or cardiovascular death was not analyzed because the nature of the data source did not allow their occurrence to be accurately identified”. This should be explained better and included in the limitations section. In fact, I don’t understand why the AA decided to exclude all-cause death in the MACE calculation. I know that is difficult to recognise real cardiovascular death, but at least all-cause death must be included in the MACE definition. 

Response: We acknowledge the validity of the Reviewer's comment. As it has been pointed out, our data extracted from EHR through NLP and ML makes too difficult recognize real cardiovascular death due to our methodology cannot infer causality. Moreover, it is worth noting that the study included a hospital population, and the detection of all-cause mortality could be underestimated given some intrinsic limitations. In this regard, we can only detect those deaths that are reflected in the free text of the EHRs, but usually deaths can occur outside the hospital, and not always be reflected on the EHRs by the healthcare professionals. Because of that, we decided to exclude both types of deaths (all-cause and cardiovascular) from the MACE definition.

Also: Why the AA excluded patients with anticoagulant treatment?

Response:  We thank the Reviewer for these thoughts. We introduced anticoagulant treatment as an exclusion criterion, trying to optimize the inclusion of incident CAD patients during the study period. By excluding patients anticoagulated before the first mention of CAD (along with some other exclusion criteria), we ensure the mention corresponds to a first diagnosis, which is needed to evaluate some of the outcomes described, and specifically, those related to antiplatelet treatment. However, anticoagulation treatments were assessed during the follow up and are shown in Table 2.

Why they performed a cross-sectional analysis stratified according to age (<65 years, 65-75 years and >75 years)? Please explain in detail these methods' criteria

Response: We have selected the mentioned age groups for two main reasons. First, they divide the study population into three balanced groups. Second, it makes our results more easily comparable to other published studies (1). We have included these details in the new manuscript version, in the methods section as required.

(1) Blin P., Darmon P., Henry P., Guiard E., Bernard M.A., Dureau-Pournin C., Maizi H., Thomas-Delecourt F., Lassalle R., Droz-Perroteau C., et al. Patients with stable coronary artery disease and type 2 diabetes but without prior myocardial infarction or stroke and THEMIS-like patients: real-world prevalence and risk of major outcomes from the SNDS French nationwide claims database. Cardiovasc Diabetol. 2021, 20, 229

Results and Discussion:

Table 1 is quite difficult to read. I suggest splitting this table. The first one only includes the three groups with a single p-value and a Bonferroni correction. The second table for the OR and linear regression. The same for Table 2.

Response:  We agree with the reviewer that Tables 1 and 2 may be challenging to read, as they contain much information. While we appreciate the advice to split the information into smaller tables, we believe that it is best to keep all data concerning a specific variable within the same table. Instead, we propose to split the tables by type of variable. We think that these changes could help the readability and the interpretability of the table content.

Explain this finding in the discussion section: “No significant differences in the relative risks of MI, unstable angina and urgent revascularisation were observed between the different age groups.” I suggest that the diagnosis and treatment of CAD are difficult, especially in patients with T2DM and older age (https://doi.org/10.1016/j.jacc.2019.07.083; 10.1016/j.ijcard.2022.07.038; http://dx.doi.org/10.1016/j.jcmg.2016.01.039; http://dx.doi.org/10.1016/j.jcmg.2014.08.001; 10.1093/ehjci/jead046). Moreover, patients with T2DM could have different phenotypes of chronic coronary syndrome with different outcomes, including those without significant CAD and microvascular dysfunction (10.1016/j.jcmg.2022.12.029; 10.2459/JCM.0000000000001305; https://doi.org/10.1016/j.jacc.2021.07.042).

Response: We thank the Reviewer for their suggestion. We agree that CAD diagnosis in diabetic elderly is still currently a challenge for the physicians and that our lack of differences in the relative risks of MI, unstable angina, or urgent revascularization between the different age groups in our study could be related to that. We have introduced new sentences in the discussion section to explain the rationale suggested by the Reviewer.

It could be useful to perform a multivariable COX analysis including age, different treatments (focus on Aspirin) and comorbidities to search independent predictors of MACE. Furthermore, a comparison in terms of MACE between T2DM CAD and NoCAD patients could be very interesting.

Response:  We thank the Reviewer for these suggestions. The present study was framed within the ACORDE project (Assessment of medical management in Coronary Diabetic Type 2 patients at high risk of cardiovascular events), which aimed to deeply describe the population with T2DM and CAD in Spain identifying the potential factors associated with the development of MACE using unstructured information in electronic health records (EHRs). In this study, we specifically aimed to provide real-world data on the age-related clinical characteristics, treatment management, and incidence of major cardiovascular outcomes in T2DM-CAD patients. However, within the ACORDE project, we have recently published the suggested analysis regarding MACE associated risk factors in the same cohort of patients (2).

Unfortunately, the comparison between CAD and Non-CAD patients with T2DM, felt out of scope for this study and Non-CAD patients were not included in the study objectives, which were approved by the Institutional Review Board of each participating site. Then, they were not included as a study population. However, we take note of this suggestion for future studies, as it could be very useful to better understanding the management and outcome of these patients.

(2). González-Juanatey C., Anguita-Sánchez M., Barrios V., Núñez-Gil I., Gómez-Doblas J.J., García-Moll X., Lafuente-Gormaz C., Rollán-Gómez M.J., Peral-Disdier V., Martínez-Dolz L., et al. Major Adverse Cardiovascular Events in Coronary Type 2 Diabetic Patients: Identification of Associated Factors Using Electronic Health Records and Natural Language Processing. J Clin Med. 2022, 11.

Please improve the Figures. Why there is a different number at risk at time 0 between figures 2 and 3?

Response:  We will include high-quality images as separate files for the figures in the final submission. Concerning the number of patients at risk at time 0, the values are 1260 (<65), 1393 (65-75), and 1419 (>75) for each age group in all the panels in both figure 2 and 3.

Minor suggestion: 

Please correct line 103: “and references 103 therein).”

Response: What the reviewer comment may have been an issue during the generation of the document by the electronic platform. What appears in the text is “([19] and references therein)” and it means that the previous content is cited in reference 19 and also, in the references included in this manuscript.

I suggest improving the fluidity and clarity of the discussion. In addition, a double check of English and abbreviations is suggested.

Response:  We thank the Reviewer for this comment. We have reviewed the writing and hope all issues have been addressed.

Round 2

Reviewer 1 Report

-All the figures are missing and tables 5 and 6 are not mentioned in the manuscript.

The English language should be checked against British english because there are some words with this grammar. 

Author Response

Dear Reviewer,

We would like to thank you for the opportunity to revise our manuscript entitled "Impact of advanced age on the incidence of major adverse cardiovascular events in patients with type 2 diabetes mellitus and stable coronary artery disease in a real-world setting in Spain".

Based on your comments, we have implemented some changes to the manuscript to further improve its content and structure. We address them below and provide a point-by-point answer.

We also provide a version of the manuscript using tracked changes and a clean version in word format.

We hope that our manuscript is now suitable for publication in JCM.

Yours sincerely, on behalf of all authors,

Dr González-Juanatey

Reviewer 1:

All the figures are missing and tables 5 and 6 are not mentioned in the manuscript.

Response: Figures have been uploaded independently to ensure the quality of the images. In the new version all the Tables are mentioned in the main manuscript.

Reviewer 3 Report

The authors answered my questions thoroughly. I believe that the manuscript has improved considerably.

Minor English revision is recommended.

Author Response

Dear Reviewer,

We would like to thank you accepting our manuscript entitled "Impact of advanced age on the incidence of major adverse cardiovascular events in patients with type 2 diabetes mellitus and stable coronary artery disease in a real-world setting in Spain".

Yours sincerely, on behalf of all authors,

Dr González-Juanatey

Reviewer 3:

The authors answered my questions thoroughly. I believe that the manuscript has improved considerably.

Response: We thank the Reviewer for their time in reviewing the manuscript and for helping us to improve it.